# Risk, Diagnostic and Predictor Factors for Classical Hodgkin Lymphoma in HIV-1-Infected Individuals: Role of Plasma Exosome-Derived miR-20a and miR-21

**DOI:** 10.3390/jcm9030760

**Published:** 2020-03-11

**Authors:** Francisco J. Hernández-Walias, Esther Vázquez, Yolanda Pacheco, José M. Rodríguez-Fernández, María J. Pérez-Elías, Fernando Dronda, José L. Casado, Ana Moreno, José M. Hermida, Carmen Quereda, Asunción Hernando, Francisco Tejerina-Picado, Víctor Asensi, María J. Galindo, Manuel Leal, Santiago Moreno, Alejandro Vallejo

**Affiliations:** 1Laboratory of Immunovirology, Infectious Diseases Department, Health Research Institute Ramon y Cajal (IRyCIS), Ramon y Cajal University Hospital, 28034 Madrid, Spain; hwfcojavier32@hotmail.com (F.J.H.-W.); esvazga@gmail.com (E.V.); mjperez90@gmail.com (M.J.P.-E.); fdronda.hrc@salud.madrid.org (F.D.); jose.casado@salud.madrid.org (J.L.C.); amorenoz@salud.madrid.org (A.M.); josemaherm@gmail.com (J.M.H.); cqueredar.hrc@salud.madrid.org (C.Q.); smguillen@salud.madrid.org (S.M.); 2Biomedicine Institute of Seville (IBiS), University Hospital Virgen del Rocío, 41013 Seville, Spain; ypacheco-ibis@us.es (Y.P.); mleal@telefonica.net (M.L.); 3Department of Internal Medicine, Gómez Ulla Central Hospital, 28047 Madrid, Spain; chemixx@gmail.com; 4Department of Medicine, 12 de Octubre University Hospital, Universidad European University of Madrid, Instituto de Investigación Hospital 12 de Octubre (imas12), 28041 Madrid, Spain; masuncion.hernando@universidadeuropea.es; 5Gregorio Marañón University General Hospital, 28007 Madrid, Spain; pacotejerina@gmail.com; 6Infectious Diseases Department, Central University Hospital of Asturias, University Medical School, 33011 Oviedo, Spain; vasensia@gmail.com; 7Group of Translational Research in Infectious Diseases, Instituto de Investigación Sanitaria del Principado de Asturias (ISPA), 33011 Oviedo, Spain; 8Clinic University Hospital of Valencia, 46010 Valencia, Spain; galindo.pepa1@gmail.com; 9Department of Internal Medicine and Infectious Diseases, Viamed Hospital, Santa Ángela de la Cruz, 41014 Seville, Spain

**Keywords:** exosome miRs, Hodgkin lymphoma, HIV-1, biomarkers, cell phenotyping

## Abstract

The incidence of classical Hodgkin lymphoma (cHL) in the HIV-1 setting has increased 5–25-fold compared to that observed in the general population. This study aimed to determine whether selected micro RNAs (miRs) and other soluble biomarkers and cellular subsets are dysregulated in cHL and could be used as biomarkers. This was a retrospective and longitudinal matched case-control study of 111 Caucasian, HIV-1-infected adult individuals, including 37 individuals with cHL and 74 with no type of cancer. Immunovirological data, plasma exosome-derived miR-16, miR-20a, miR-21, miR-221, miR-223, miR-106a, miR-185, miR-23, miR-30d, miR-222, miR-146a and miR-324, plasma IL-6, sCD14, sCD27, sCD30, sIL-2R, TNFR1, and cell phenotyping of T and B lymphocytes and natural killer (NK) cells were analyzed. Before cHL diagnosis, miR-20a, miR-21, and sCD30 were higher in cHL (*p* = 0.008, *p* = 0.009 and *p* = 0.042, respectively), while miR-16 was down-regulated (*p* = 0.040). miR-20a and miR-21 were independently associated with cHL (*p* = 0.049 and *p* = 0.035, respectively). The combination of miR-20a and miR-21 showed a good AUC value of 0.832 with a moderate likelihood ratio positive (LR+) value of 5.6 and a slight likelihood ratio negative (LR−) value of 0.23. At cHL diagnosis, miR-20a, miR-21 and miR-324 were overexpressed in cHL (*p* = 0.005, *p* = 0.024, and *p* = 0.001, respectively), while miR-223, miR-16, miR-185 and miR-106a were down regulated (*p* = 0.042, *p* = 0.007, *p* = 0.006, and *p* = 0.002, respectively). In addition, sCD14, sCD27, sCD30 and IL2R levels were higher in these individuals (*p* = 0.038, *p* = 0.010, *p* = 0.030, *p* = 0.006, respectively). miR-20a was independently associated with cHL (*p* = 0.011). The diagnostic value of miR-20a showed good AUC value of 0.754 (*p* = 0.074) with a slight LR+ value of 2 and a slight LR− of 0.25. After chemotherapy, miR-20a was higher in those individuals who had an adverse outcome (*p* < 0.001), while sCD14 and sCD30 were higher (*p* < 0.001). A specific signature of miRs and cytokines associated with a subsequent cHL diagnosis was found in this study, especially miR-20a and miR-21. Also, another biomarker signature was found at cHL diagnosis, with a relevant discriminant disease value for miR-20a. Of note, miR-20a expression was higher in those individuals who had an adverse clinical outcome after chemotherapy.

## 1. Introduction

Combination antiretroviral therapy (cART) has improved immune function and reduced the risk of developing AIDS in individuals with HIV-1 infection [1,2]. In line with the improved viral control achieved by cART, the risk of developing AIDS-defining cancers, such as Kaposi sarcoma (KS) and non-Hodgkin lymphoma (NHL), have declined significantly, whereas non-AIDS defining malignancies, such as plasmablastic lymphoma (PBL) and classical Hodgkin´s lymphoma (cHL), have remained stable or increased in incidence [3,4,5,6].

The incidence of cHL in the HIV setting have increased 5–25-fold, suggesting that despite suppressive cART, the ongoing immune suppression (driven by ongoing HIV replication among other factors) and the aged population have contributed to this rise [7,8,9,10,11,12,13]. On the other hand, individuals with severe immune suppression have a lower cHL incidence, suggesting that the improvement of CD4 counts associated to cART and the rise of immune competence may explain the observed increase of cHL incidence [14,15,16,17,18,19]. Moreover, the incidence of cHL among HIV-1-infected individuals has been estimated to be 10–20-fold higher than in the general population [7,8,9,10,11,12,13]. Moreover, nearly all cases of cHL in individuals with HIV infection are associated with EBV infection, in contrast to only 20–50% of HIV-negative individuals. Clinical observations have reported that moderate immune activation or reconstitution after cART may increase B-cell stimulation, as well as the burden of EBV-infected lymphocytes and the consequent risk of cHL development [20,21].

The most common cHL subtypes in HIV-positive individuals are mixed cellularity and lymphocyte depleted subtypes, which are the most unfavorable histological subtypes. In severe immune suppression, the nodular sclerosis subtype becomes infrequent [22]. Despite the increased incidence of cHL in the cART setting, significant improved outcomes have been observed after treatment, including decreased morbidity and mortality [23]. Hence, individuals with HIV-cHL show a similar overall response and survival rates to their matched non-HIV countertypes [24,25,26].

Micro RNAs (miRs) reside inside exosome vesicles and are released by cells and present in all body fluids. They have been largely studied in cancer processes as biomarkers that might influence the immune system [27,28]. Both disease presence and progression have been associated with an increase in exosome release and their molecular content. These molecules could influence the homeostasis cell balance, promoting hematopoietic stem cells and regulate the immune system by modifying the levels of soluble cytokines [29]. Furthermore, miRs have a role in the pathogenesis of HIV-1 by repressing the translation of HIV-1 proteins located in resting CD4 cells that contribute to the latency of the virus among other mechanisms [30,31,32,33,34].

This study was performed to analyze whether selected miRs and other soluble biomarkers are dysregulated and could be used as biomarkers with a risk value for cHL development in HIV-1-infected individuals (prediagnostic samples). This would serve to discriminate this disease from other disease types (diagnostic samples) and to predict cancer treatment outcomes (post-treatment samples). The association of the expression of these biomarkers and immune cell subsets was also examined.

## 2. Materials and Methods

### 2.1. Study Design and Population

A retrospective and longitudinal matched case-control study was performed of 111 Caucasian HIV-1-infected adults provided by the HIV BioBank integrated in the Spanish AIDS Research Network Cohort (CoRIS), the HIV Cohort of the Ramón y Cajal University Hospital (Madrid, Spain) [35,36] and the HIV Cohort of the Virgen del Rocío University Hospital (Seville, Spain). Samples were processed following current procedures and frozen immediately after their reception. Thirty-seven individuals who were diagnosed with classical Hodgkin lymphoma from 2000 to 2012 (cHL group) and 74 individuals who were not diagnosed with cHL or other kinds of cancer (Control group), were included in the study. The control group was selected with 1:2 ratio adjusted by age, gender, HIV-1 transmission, HCV infection, and antiretroviral treatment to reduce selection bias and improve internal validity. Plasma samples were available in 15 patients before cHL diagnosis, 25 patients at cHL diagnosis, and 18 patients after treatment, while PBMC were available in six patients before cHL diagnosis, 9 patients at cHL diagnosis, and 9 patients after treatment (Appendix A).

All individuals with a cHL diagnosis had been treated with chemotherapy, usually six cycles of doxorubicin, bleomycin, vinblastine, and dacarbazine (ABVD). The study was approved by the Ethics Committee for Clinical Research of the respective institutions and performed according to the principles of the Declaration of Helsinki. Written informed consent was obtained from all the participants.

### 2.2. Laboratory Measurements

CD4 and CD8 counts were determined in fresh blood by flow cytometry. Plasma HIV-1 RNA quantification was measured by quantitative polymerase chain reaction (qPCR, COBAS Ampliprep/COBAS Taqman HIV-1 test, Roche Molecular Systems, Basel, Switzerland) according to the manufacturer’s instructions, with a detection limit of 40 HIV-1 RNA copies/mL. HCV antibodies were assayed by EIA (Siemens Healthcare Diagnosis, Malvern, Pennsylvania) and plasma HCV RNA quantification by RT-qPCR (COBAS Amplicor, Roche Diagnosis, Barcelona, Spain). 

### 2.3. Quantification of Plasma Exosome-Derived miRs

Cryopreserved EDTA-plasma samples were thawed and sequentially centrifuged to remove cell debris, as described previously [37]. They were subsequently treated with thrombin and DNase to prevent platelet and DNA contamination, respectively, and filtered to eliminate larger vesicles such as large extracellular vesicles and apoptotic bodies. Exosomes were precipitated using the miRCURY Exosome isolation kit (Exiqon A/S, Vedbaek, Denmark) and quantified for exosome content by the ExoELISA-ULTRA CD63 assay (SBI System Bioscience, Mountain View, CA, USA).

RNA from isolated exosome-enriched pellets was extracted using the miRCURY RNA isolation kit-Biofluids (Exiqon A/S) and UniSp2-4-5 RNA templates were added as an internal control. The concentration and purity of eluted RNA were analyzed with the NanoDrop instrument (Thermo Scientific, Waltham, MA, USA). All RNA samples with 260/280 ratio between 1.8 to 2.0 and 260/230 ratio near 2.0 were considered suitable for further analyses. Ten nanograms of RNA was reverse transcribed in 15 µl reactions (miRCURY LNA Universal RT microRNA PCR, Exiqon A/S), including the UniSp6 RNA spike-in template reaction control. cDNA was used for PCR reactions in triplicate using the ExiLENT SYBR Green Master Mix (Exiqon A/S). LNA-based primers (Exicon A/S) for hsa-miR-451a and hsa-miR-23a were assayed to detect levels of hemolysis. Primers for hsa-miR-103a-3p, hsa-miR-425-5p, and hsa-miR-93-5p were quantified as reference controls for the calculation of the relative concentration of the following miRs: hsa-miR-16-5p, hsa-miR-20a-5p, hsa-miR-21-5p, hsa-miR-221-3p, hsa-miR-223-3p, hsa-miR-106a-5p, hsa-miR-185-5p, hsa-miR-23-5p, hsa-miR-30d-5p, hsa-miR-222-3p, hsa-miR-146a-5p, and hsa-miR-324-3p.

Thermocycler conditions included a hot start at 95 °C/10 min followed by 40 cycles of 95 °C/15 s and 60 °C/45 s using the LyghtCycler 480 II instrument (Roche, Basel, Switzerland). Amplification curves were analyzed using the Roche LightCycler 480 version 1.5.1.62 software. Reaction specificity was ascertained by the melt curve procedure. Ct values >35 and a standard deviation between triplicates >0.3 Ct were considered unreliable and were excluded from further analysis. The expression levels of single miRs relative to the mean reference miRs expression were calculated using the ΔCt method (Ct target miR minus mean references Ct) and are shown as log_2_ 2^−ΔCt^ [38,39,40].

### 2.4. Soluble Plasma Cytokine Quantification

Plasma samples were used to quantify the following markers of inflammation and immune activation: interleukin-6 (IL-6), soluble CD14 (sCD14), soluble interleukin-2 receptor (sIL-2R), tumor necrosis factor receptor type 1 (TNFR1), sCD27, and sCD30 using the R&D Luminex HS assay (R&D Systems, MN, USA). Human Procarta-Plex immunoassays (ThermoFisher Scientific, Minneapolis, MN, USA) were used in combination with the Luminex instrument platform (MagPix, Luminex Corporation, Waltham, MA, USA), according to the manufacturer´s instructions.

### 2.5. Cellular Phenotyping: Antibody Staining, Flow Cytometry Acquisition, and Analysis

Cryopreserved peripheral blood mononuclear cells (PBMCs) were thawed and counted. Briefly, 5 × 10^5^ cells were incubated with selected antibodies and stained for viability (Miltenyi Biotech, Madrid, Spain) for 20 min at 4 °C, washed and resuspended in PBS containing 1% azide. Flow cytometry was performed using a Gallios flow cytometer and data analysis was performed using the Kaluza software (Beckman-Coulter, Brea, CA, USA). Isotype controls were also used in the analysis. The flow cytometry gating strategies for the analysis of the different cell subsets are shown in Appendix A.

### 2.6. B lymphocyte Phenotyping and Activation

After initially gating lymphocytes according to morphological parameters, at least 50,000 CD19^+^/CD3^-^ cells stained using anti-CD19-APC (clone LT19) and anti-CD3-VioBlue (clone BW264/56), were selected and tested for cell viability for further analysis. Monoclonal antibodies used for B cell phenotyping included: anti-CD20-PerCP (clone LT20), anti-CD10-FITC (clone 97C5), anti-CD27-PE Vio770 (clone M-T271), anti-CD21-PE (clone HB5) and anti-IgG-APC Vio770 (clone IS11-3B2.2.3). Cell subpopulations analyzed were: immature/transitional (CD10^+^/CD27^−^), naïve (CD21^hi^/CD27^−^), resting memory (CD21^hi^/CD27^+^), tissue-like memory (CD21^lo^/CD27^−^), activated memory (CD21^lo^/CD27^+^ and plasmablast (CD20^-^/CD21^lo^/CD27^++^).

### 2.7. T Lymphocyte Phenotyping and Activation

After initially gating lymphocytes according to morphological parameters, at least 50,000 CD3^+^ T cells stained using anti-CD3-VioBlue (clone BW264/56), were selected and tested for cell viability for further analysis using the following monoclonal antibodies: anti-CD4-FITC (clone M-T466), anti-CD8-APC R700 (BD Horizon, clone RPA-T8), anti-CD38-APC (clone IB6), anti-HLA-DR-APC Vio770 (clone AC122), anti-CD45RA-PerCP (clone T6D11), anti-CCR7-PE Vio770 (clone REA546) and anti-PD1-PE (clone REA1165). Activated CD4/CD8 T cells were defined by the co-expression of CD38 and HLA-DR. Subpopulations were defined as: naive, CD3^+^CD4^+^(CD8^+^)CD45RA^+^CCR7^+^; effector memory (EM) CD3^+^CD4^+^(CD8^+^)CD45RA^-^CCR7^−^; central memory (CM) CD3^+^CD4^+^(CD8^+^)CD45RA^-^CCR7^+^; and transitional memory (TemRA) CD3^+^CD4^+^(CD8^+^)CD45RA^+^CCR7^−^.

### 2.8. Natural Killer Cell Phenotyping and Activation

After the selection of single cells and CD14^−^, CD19^−^, CD3^−^ and CD4^−^ cells, the monoclonal antibodies anti-CD56-FITC and CD16-APC-Vio770 were used to define the following subpopulations: CD56^dim^CD16^−^, CD56^br^CD16^−^, CD56^br^CD16^+^, CD56^dim^CD16^+^ and CD56^−^CD16^+^. Furthermore, anti-CD94-PerCP, anti-NKp46-PE, anti-NKp30-PE-Vio770 and anti-NKG2D-APC were used to define inhibitory and activating NK cells.

## 3. Statistical Analyses

Continuous variables were expressed as the median and interquartile range (IQ_25–75_) and categorical variables by frequencies and proportions. The Mann–Whitney *U* test (non-parametric) for independent samples was used to compare continuous variables. Differences between categorical variables were evaluated using contingency tables (Chi-square distribution). Univariate logistic regression analysis was used for the categorical dependent variables (cHL versus not-cHL) using all the variables studied. Variables with *p* < 0.1 according to the univariate analysis were included in the conditional logistic regression analysis to estimate the regression coefficient (β) and 95% confidence interval for the association of each variable with cHL risk. These conditional logistic models were adjusted for age, gender, HIV-1 transmission, HCV infection, and antiretroviral treatment. CD4 and CD8 counts were not introduced in the conditional logistic regression model because of their collinearity with the CD4/CD8 ratio. Variables with *p* < 0.05 were independently associated with cHL. The Wilcoxon signed-rank test was used to compare paired-samples to analyze the evolution of the biomarkers. Spearman’s rank correlation coefficient was used to measure the association between two variables. In addition, receiver operating characteristic (ROC) curve analysis was performed to evaluate the discrimination threshold (with 95% confidence intervals) of the variables. This was performed via their sensitivity/specificity pair and the area under the curve (AUC) that measured the accuracy of a diagnostic test to discriminate between two diagnostic groups. In addition, the likelihood-ratio (LR) test was performed to assess how these variables increase (LR+) or decrease (LR−) the probability to develop cHL. Statistical analysis was performed using the SPSS software 22.0 (SPSS Inc., Chicago, IL, USA).

## 4. Results

The immunovirologic characteristics of the 111 HIV-1-infected individuals of the study are shown in Table 1. These included 37 individuals with classical Hodgkin lymphoma (cHL group) and 74 individuals without any kind of neoplasia (control group). Variables that were adjusted between the cHL group and the control group included age, gender, HIV-1 transmission route, HCV infection status, and HIV-1 treatment. Patients with cHL were further analyzed a median of 11 [5–28] months before cHL diagnosis (Pre-cHL) and a median of 13 [10–22] months after chemotherapy. The pathological stage and classification of the cHL are described in Table 2. Biological samples available from individuals in the cHL group and control are summarized in detail in Appendix A.

### 4.1. Biomarkers Profile at Pre-cHL Diagnosis

Before cHL diagnosis, the CD4 count in the cHL group (378 [202–545] cells/mL) was higher compared to the levels at cHL diagnosis (259 [115–385] cells/mL), as shown in Table 1. Although there was no statistical significance (*p* = 0.358). Furthermore, these levels were lower compared to the control group (511 [285–695] cell/mL), although not statistically significant (*p* = 0.142). No data on CD8 count was available at this point.

The levels of miR-20a and miR-21 were higher compared to the control group (*p* = 0.008 and *p* = 0.009, respectively) and similar to those at cHL diagnosis. Whereas, the level of miR-16 was lower compared to the control group (*p* = 0.040) and similar to those at cHL diagnosis (Figure 1). Similar levels of the rest of the miRs were found pre-cHL diagnosis compared to the control group (*p* > 0.05 in all cases). The analysis of paired samples (Wilcoxon analysis) showed that the levels of miR-324 increased from pre-cHL to cHL time points (*p* = 0.005), while the levels of miR-185 and miR-106a decreased (*p* = 0.009 and *p* = 0.010, respectively). The rest of the miRs remained unchanged from pre-cHL to cHL time points.

The levels of most of these cytokines were similar to those found in the control group, with the exception of sCD30. The levels of sCD30 were higher compared to the control group (*p* = 0.042), as shown in Figure 2. From pre-cHL to cHL time points, only the levels of sCD14 increased (Wilcoxon paired samples analysis, *p* = 0.033). On the other hand, the levels of the rest of the cytokines remained similar.

The level of the CD56^br^CD16^−^ NK cell subset was lower, which almost reached statistical significance (*p* = 0.050), as shown in Table 3. All activating markers for NK cells in the cHL group were similar to those found in the control group. Similar levels of B lymphocyte, CD4 T cell, and CD8 T cell subsets were found compared to the control group. CD8 cell activation in the cHL group was higher compared to the control group, although this was not statistically significant (*p* = 0.050).

Univariate analysis showed that only miR-16, miR-20a, miR-21, sCD27, and sCD30 were associated with cHL before cHL diagnosis, as shown in Table 4. Interestingly, miR-20a directly correlated with miR-21 (*p* = 0.016, r = 0.399) and inversely with miR-16 (*p* = 0.027, r = −0.368). In addition, miR-21 directly correlated with sCD27 (*p* = 0.010, r = 0.424) and sCD30 directly correlated with sCD27 (*p* = 0.035, r = 0.353). Multivariate analysis performed with these variables showed an independent association of miR-20a and miR-21 with cHL (*p* = 0.049 and *p* = 0.035, respectively), as shown in Table 4. The ROC curve analysis showed a good AUC value for miR-20a of 0.762 (*p* = 0.008) with a slight LR+ value of 2.19 and moderate LR− value of 0.11. In addition, the ROC curve analysis showed a good AUC value for miR-21 of 0.756 (*p* = 0.010) with a slight LR+ value of 2.33 and a slight LR− value of 0.47, as shown in Table 5. The combination of miR-20a and miR-21 showed a good AUC value of 0.832 (*p* = 0.001) with a moderate LR+ value of 5.6 and a slight LR− value of 0.23.

### 4.2. Biomarkers Profile at cHL Diagnosis

The cHL group had lower CD4 and CD8 counts compared to controls (*p* < 0.001 and *p* = 0.018, respectively), while their respective percentages were similar. In addition, the CD4/CD8 ratio was lower in the cHL group compared to the control group (*p* = 0.046), as shown in Table 1. Pre-ART HIV-1 RNA was similar in both groups (*p* = 0.677), while the nadir CD4 count in the cHL group was lower compared to the control group, although there was no statistical significance (*p* = 0.119). The time from HIV-1 diagnosis to cHL diagnosis or sample collection was higher in the control group (*p* = 0.031), while the differences between the time from HIV-1 diagnosis to ART initiation or the time under ART were not statistically significant between the two groups (*p* = 0.074 and *p* = 0.816, respectively).

The levels of miR-20a, miR-21, and miR-324 were higher in the cHL group compared to the levels found in the control group (*p* = 0.005, *p* = 0.024, and *p* = 0.001, respectively), while the levels of miR-223, miR-16, miR-185, and miR-106a were lower (*p* = 0.042, *p* = 0.007, *p* = 0.006, and *p* = 0.002, respectively), as shown in Figure 1. On the other hand, the levels of miR-23, miR-30d, miR-146a, miR-221 and miR-222 were similar in both groups (*p* > 0.05 in all cases). The levels of sCD14, sCD27, sCD30, and IL2R were higher in the cHL group (*p* = 0.038, *p* = 0.010, *p* = 0.030, *p* = 0.006, respectively), while IL-6 and TNFR1 were similar (*p* = 0.432 and *p* = 0.614, respectively), compared to the control group (Figure 2). Interestingly, grouping individuals according to their cHL classification into two groups, those with mixed cellularity/lymphocyte depleted, and those with nodular sclerosis/lymphocyte-rich, higher levels of miR-324, sCD30, and sIL2R were found in individuals with mixed cellularity/lymphocyte depleted classification (*p* = 0.020, *p* = 0.001, and *p* = 0.001, respectively).

The level of CD56^br^CD16^−^ NK cell subset in the cHL group was lower compared to the control group (*p* = 0.003), while the level of CD56^br^CD16^+^ NK cell subset was higher in the cHL group (*p* = 0.010), as shown in Table 3. The inhibitory CD94^+^ cell subset was higher (*p* = 0.040) and the activating NKp46^+^ cell subset was lower (*p* = 0.031) in the cHL group compared to the levels found in the control group. The level of resting memory B lymphocytes was higher (*p* = 0.048) and the level of activated memory B lymphocytes was lower (*p* = 0.018) in the cHL group compared to the levels found in the control group. The level of the central memory CD4 T cell subset was higher while the level of the effector memory CD4 T cell subset was lower in the cHL group compared to the control group, although only with a trend towards statistical significance (*p* = 0.072 and *p* = 0.050, respectively). The central memory CD8 T subset was higher and the effector memory CD8 T cell subset was lower in the cHL group compared to the control group (*p* = 0.006 and *p* = 0.011, respectively). In addition, CD8 T cell activation was higher in the cHL group compared to the control group (*p* = 0.031).

Univariate analysis showed that nadir CD4 count, CD4 count, CD8 count, CD4/CD8 ratio, miR-20a, miR-16, miR-106a, miR-21, miR-324, miR-185, miR-223, sCD14, sCD27, sCD30 and sIL2R were associated with cHL, as shown in Table 4. The Spearman’s rank correlation coefficient test among variables is shown in detail in Appendix A. Briefly, CD4 nadir directly correlated with both CD4 count and CD4/CD8 ratio but no correlation with either miRs or cytokines was found. CD4 count directly correlated with miR-16 and miR-106a and inversely with miR-324, while the CD4/CD8 ratio inversely correlated with both miR-20a and miR-324. On the other hand, some miRs showed a statistically significant correlation with cytokines. This included miR-20a that directly correlated with sCD27, miR-16 and miR-223 that inversely correlated with sCD30, miR-21 that directly correlated with sCD14 and sCD27, and miR-185 that inversely correlated with IL2R. Multivariate analysis was performed to determine which variables were independently associated with cHL by analyzing all the variables with *p* < 0.1 according to the univariate (unadjusted) analysis, as shown in Table 5. Multivariate analysis showed that only miR-20a was independently associated with cHL (*p* = 0.011). The diagnostic value analyzed by the ROC curve of miR-20a showed a good AUC value of 0.754 (*p* = 0.005) with a slight LR+ value of 2 and a slight LR− of 0.25, as shown in Table 5. Unfortunately, the predictive value of these biomarkers according to their treatment outcome could not be statistically analyzed as there were only three plasma samples at cHL diagnosis from individuals with adverse outcome, such as partial response, relapse or death, was only three.

Individuals with advanced cHL stages received 6 to 8 cycles of ABVD, while patients with early stages (stages I and II) received combined therapy, including two cycles of ABVD followed by radiotherapy and two subsequence cycles of ABVD. The level of CD4 count after chemotherapy was higher (338 [261–410] cells/mL) compared to those levels at cHL diagnosis (259 [115–385] cell/mL). Although it was not statistically significant (*p* = 0.377). No data were available on the levels of CD8 count. Out of the 21 individuals with a cHL diagnosis and plasma sample collected before chemotherapy, 15 had a complete response to chemotherapy, one had a partial response, one relapsed, and four had a cHL-related death.

A trend to reach the levels of those found in the control group was found for all miRs, although the levels remained higher in the case of miR-20a, miR-106a (*p* = 0.031 and *p* = 0.029, respectively), as shown in Figure 1. The levels of miR-223, miR-185, and miR-106a increased after cHL treatment (Wilcoxon test, *p* = 0.001, *p* = 0.010, and *p* = 0.002, respectively), while the level of miR-21 decreased (Wilcoxon test, *p* = 0.002) compared to the levels found at cHL diagnosis.

The levels of sCD14 and sCD30 were still higher compared to the control group (*p* < 0.001 in both cases), while the level of IL-6 was lower (*p* = 0.003). The level of the remaining cytokines reached similar values compared to the control group. The changes in the levels of the cytokines were not statistically significant (Wilcoxon paired samples analysis). Interestingly, two out of the three individuals with an adverse outcome, such as a partial response, relapse, or death, had elevated levels of sIL2R (greater than the optimal cutoff value of disease-free survival that was 4.23 pg/mL, data not shown).

The CD56^br^CD16^+^ NK cell subset remained higher (*p* = 0.019) in the cHL group compared to the control group. Only the level of activating NKp46^+^ in the cHL group remained lower (*p* = 0.012) compared to the levels found in the control group. No modification of the levels of the rest of the cell subsets was found, with the exception of the level of central memory CD8 T subset that was still higher compared to the control group (*p* = 0.019), as shown in Table 4.

Biomarkers could be analyzed in 18 individuals after chemotherapy. Interestingly, the level of miR-20a was significantly higher in those with an adverse outcome (*p* = 0.003), either partial response, relapse, or death, compared to those who had a complete response. However, the number of individuals with an adverse outcome was very low (four individuals) (Figure 3). 

## 5. Discussion

The initial diagnosis of classical Hodgkin lymphoma (cHL) requires tissue sampling. However, early detection of non-invasive disease biomarkers with a risk value for cHL in the setting of HIV-1 infection can aid a diagnosis prior to clinically detectable evidence of the disease. Thus, they are of great clinical interest. Many studies of circulating miRs in cancer demonstrated a risk value for cancer even at early stages, as well as the prognostic potential using a combination of miR expression levels and other biomarkers [41]. These short RNAs may be extracted from blood in a non-invasive way, they are robust, survive treatment and endure prolonged storage conditions. Several studies involving the role of miRs in cHL have been reported [42,43,44,45,46,47,48], but very few in the context of an HIV-1 infection [49].

In this study, we found plasma exosome-derived miR-20a and miR-21 to be up-regulated one year before a cHL diagnosis, compared to controls. In addition, the level of sCD30, a B-cell stimulation marker, was also higher in these individuals. This is in accordance with previous reports in cHL [50] and not-cHL [51]. On the other hand, the down-regulation of miR-16 was also observed in the pre-cHL stage. In contrast with our study, Levin et al. [50] found elevated serum levels of IL-6 prior to cHL diagnosis, although in individuals without HIV-1 infection. After multivariate analysis, both miR-20a and miR-21 were found to be independently associated with a future cHL diagnosis. Interestingly, miR-21 correlated directly with both miR-20a and sCD27. This has been reported to induce B cell activation and to be a good biomarker for HIV-related lymphomas [49], which is in agreement with our results. Although sCD27 and sCD30 play an important role in regulating cellular activity in subsets of T, B, and natural killer cells [52], no differences in these subsets were found in pre-cHL diagnosis individuals compared to controls in this study. Although, there was a good direct correlation between both markers. Despite the lack of statistical significance, elevated T cell activation and low NK activating markers have been found in pre-cHL individuals. In addition, the CD56^br^CD16^−^ NK cell subset was decreased in individuals prior to cHL. This has been found to play an immune-regulatory role via chemokine and cytokine secretions, which attract and activate antitumor cells from both innate and adaptive arms of the immune system [53,54,55]. Interestingly, our data revealed that the combination of miR-20a and miR-21 had a good value for cHL risk, with good sensitivity and specificity (80.0% and 85.7%, respectively). A moderate positive probability and a slight negative probability to develop cHL (likelihood ratio) were also found. Therefore, they may be good plasma biomarker candidates for the risk of cHL in the HIV-1 setting.

Among individuals with diagnosed cHL in this study, a lower CD4/CD8 ratio was found compared to controls. This is in agreement with other publications that describe a decrease in CD4 and CD8 counts and the CD4/CD8 ratio in cHL, with [56] or without [57,58] HIV-1 infection. The low levels of the CD4 count and CD4/CD8 ratio directly correlated with miR-16 and miR-106a and inversely correlated with miR-324 and miR-20a. In agreement with our results, elevated expression of miR-20a has been reported in individuals with cHL not associated with HIV-1 [43,45,46]. Overexpression of miR-21 is also found in individuals with cHL [44,59,60] and in HIV and non-HIV-related lymphomas [49]. In addition, silencing miR-21 was found to be correlated with an increased clinical outcome [45]. In contrast with our results, miR-16 has been reported to be upregulated in individuals with cHL [43,46] and also with miR-223 in HIV-1 individuals compared to healthy controls [61]. We found that these overexpressed levels are higher in HIV-infected individuals who have not developed cHL, suggesting possible repression of their expression in HIV-associated cHL. Of note, miR-21 and miR-20a were associated with elevated levels of sCD27 and sCD14, both are reported to be associated with cHL [50,51]. In parallel, elevated sCD30 and sIL2R levels, reported in pre-cHL and cHL [50,51,52], were associated with repressed levels of miR-16, miR-223 and miR-185. Multivariate analysis showed that only miR-20a was independently associated with cHL, with a good discrimination value of the disease, with slightly positive and negative probabilities of discrimination.

We did not include cell subsets in the multivariate analysis since the number of individuals was very low. Nevertheless, some interesting findings were observed in this reduced group of individuals that are worth discussing. In this way, individuals with cHL had lower levels of the CD56^br^CD16^−^ NK cell subset and higher levels of the CD56^br^CD16+ NK cell subset, with an overall impairment of the NK cell activity. This was shown by an enhancement of inhibitory NK cell markers and a depletion of activating NK cell markers, as found in pre-cHL samples. The level of resting memory B lymphocytes in cHL was higher to the detriment of the level of activated memory B lymphocytes. In addition, individuals with cHL had less CD4 and CD8 effector memory T cells, contributing to the impairment of the immune response.

Despite the limited number of plasma samples at cHL diagnosis with subsequent treatment outcome, two out of three individuals with an adverse outcome had elevated sIL2R. This is in accordance with a previously reported study demonstrating that sIL2R can be a useful prognosis factor in individuals with cHL treated with chemotherapy [62,63]. On the other hand, and in contrast to our study, Marri et al. [62] found that elevated pretreatment serum levels of IL-6 and IL2R were associated with inferior survival after treatment in cHL. We did not find elevated IL-2R levels in cHL, probably due to the HIV-1 infection that drove the elevated levels in cHL individuals and in controls (both with HIV-1 infection). We also found that the expression of miR-20a was higher in those individuals who had an adverse outcome. This finding is in accordance with a previously reported study that revealed plasma levels of miR-20a to be associated with a higher hazard ratio for mortality in individuals without an HIV-1 infection [45]. We also found that miRs tended to reach the levels of those found in controls. In addition, the levels of sCD14 and sCD30 remained higher compared to the control group but decreased after chemotherapy, which is also in agreement with other studies [62,64].

It is difficult to compare our results to those in the literature since the study of the HIV-infected population is very scarce and there are many different studies analyzing biomarker expression profiles from different plasma sub-fractions (e.g., isolated EVs, size-exclusion chromatography, protein-bound, etc.). One strength of this study has is that plasma samples collected 1–2 years before and after the onset of cHL were included. This allowed the prospective analysis of the diagnostic and prognostic capacity of plasma exosome-derived miRs and soluble immune biomarker levels. The study was also able to control for potential confounding by matching the individuals that comprised the comparison groups. With regard to limitations, there was a low number of pre-chemotherapy samples with a recorded outcome to evaluate the prognosis value of the biomarkers. Furthermore, not all individuals had paired-samples, and neither non-HIV-1-infected patients with cHL nor healthy individuals were included in this study.

In conclusion, exosome-derived miRNAs are promising biomarkers for the early detection of cHL in the setting of HIV-1 infection. Future larger studies, in conjunction with global miRNA screening will be needed for the identification and verification of circulating miRNAs as biomarkers for cHL.

## Figures and Tables

**Figure 1 jcm-09-00760-f001:**
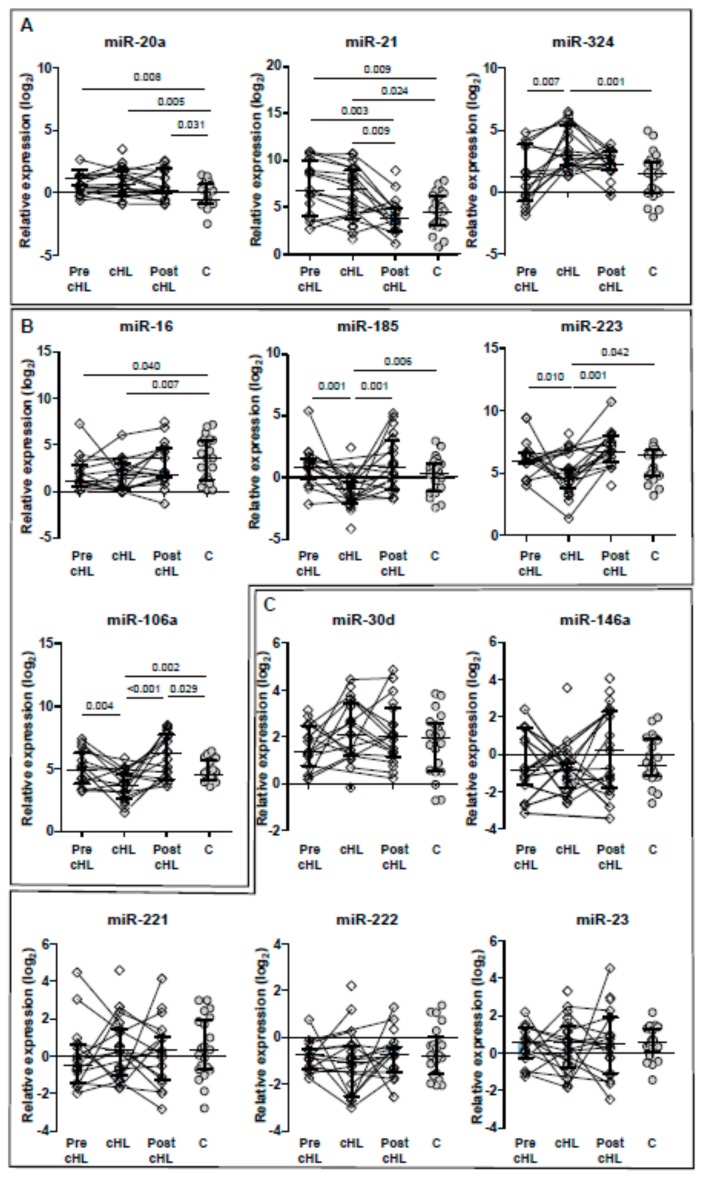
Relative expression of miRs before cHL (Pre-cHL), at cHL diagnosis (cHL) and post-chemotherapy (Pst-cHL), and in individuals with no cHL (Controls, C). A, miRs over-regulated at cHL diagnosis compared to controls; B, miRs down-regulated at cHL diagnosis compared to controls; and C, miRs with no differences in expression compared to controls. Paired-samples are represented by continuous lines. Only significant *p* < 0.05 values are shown (Mann-Whitney *U* test).

**Figure 2 jcm-09-00760-f002:**
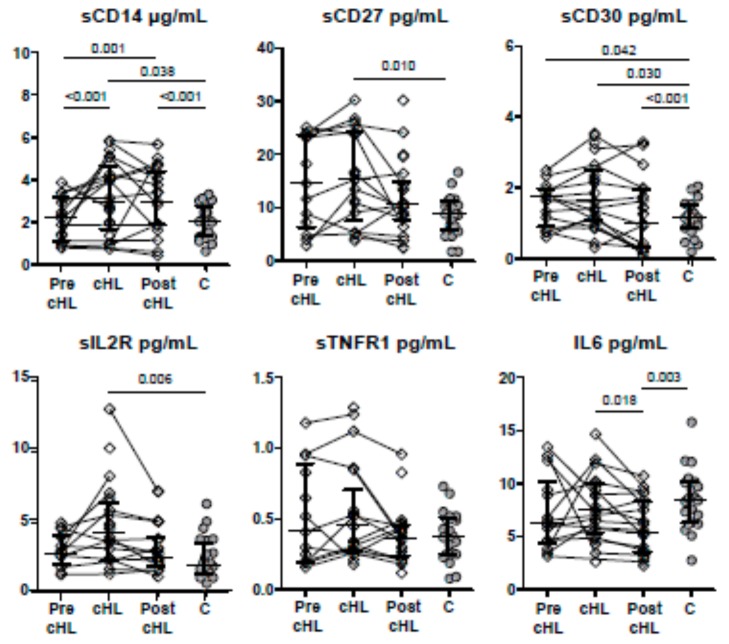
Cytokine concentration before cHL (pre-cHL), at cHL diagnosis (cHL) and post-chemotherapy (post-cHL), and in individuals with no cHL (Controls, C). Paired-samples are represented by continuous lines. Only significant *p* < 0.05 values are shown (Mann-Whitney *U* test).

**Figure 3 jcm-09-00760-f003:**
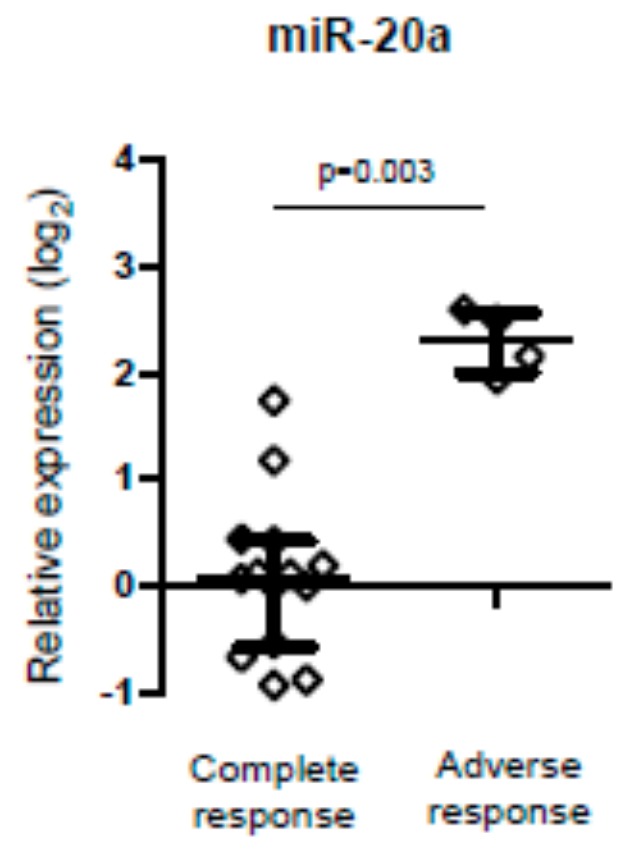
Relative expression of miR-20a after chemotherapy treatment in individuals with cHL according to clinical outcome, i.e., complete versus adverse response (Mann-Whitney *U* test).

**Table 1 jcm-09-00760-t001:** Immunovirologic characteristics of the HIV-1-infected individuals at cHL diagnosis.

	HIV-1-Infected Individuals with Classical Hodgkin Lymphoma N = 37	HIV-1-Infected Individuals (Controls) N = 74	*p*
Adjusted factors			
Age at HIV-1 diagnosis (years)	34 [29–43]	31 [26–38]	ns
Age at cHL or sample (years)	45 [38–46]	44 [39–47]	ns
Gender (female)	6 (16.2%)	12 (16.2%)	ns
HIV-1 transmission risk			
MSM	18 (48.6%)	36 (48.6%)	ns
Heterosexual	10 (27.0%)	20 (27.0%)	ns
Former injecting drug users	6 (16.2%)	12 (16.2%)	ns
Other	3 (8.1%)	6 (8.1%)	ns
Anti-HCV antibodies			
Positive	12 (32.4%)	24 (32.4%)	ns
Negative	24 (64.8%)	50 (67.6%)	
Unknown	1 (2.7%)	0	
HIV-1 treatment			
cART	28 (75.6%)	58 (76.3%)	ns
None	9 (24.3%)	18 (23.7%)	ns
HIV-1 load (RNA copies/mL) *	4.8 [3.5–5.4]	4.5 [4.2–5.1]	ns
Not adjusted factors			
Time from HIV-1 diagnosis to cHL or sample (months)	63 [16–103]	129 [52–187]	0.031
Time from HIV-1 diagnosis to ART initiation (months)	15 [1–54]	17 [3–62]	0.816
ART exposure (months)**	47 [15–126]	96 [42–158]	0.074
Pre-ART HIV-1 load (RNA copies/mL)	5.1 [4.4–5.5]	5.0 [4.5–5.5]	0.677
CD4 nadir (cells/mm3)	120 [70–241]	196 [83–292]	0.119
CD4 count (cells/mm3)	259 [115–385]	511 [285–695]	<0.001
CD4 T cell percentage	20 [13.7–30]	24.4 [17.5–30.6]	0.516
CD8 count (cells/mm3)	554 [370–892]	872 [650–1072]	0.018
CD8 T cell percentage	53.7 [32.7–60.2]	42.9 [36.8–53.5]	0.174
CD4/CD8 ratio	0.48 [0.32–0.81]	0.66 [0.47–0.87]	0.046
Pre-cHL CD4 count (cells/mm3)	378 [202–545]	515 [285–695]	0.142
Post cHL CD4 count (cells/mm3)	338 [261–410]	515 [285–695]	0.006

*, in individuals who had not received ART at the moment of cHL diagnosis or sample collection; **, time to cHL diagnosis or sample collection. Median, IQR. p, Mann-Whitney *U* test; ns, not significant. Abbreviations: cHL: classical Hodgkin lymphoma; MSM: men who have sex with men; ART: antiretroviral treatment.

**Table 2 jcm-09-00760-t002:** World Health Organization classification of classical Hodgkin lymphoma.

	N = 37
Pathologic Stage	
I	4 (10.8%)
II	3 (8.1%)
III	4 (10.8%)
IV	15 (40.5%)
Unknown	11 (29.7%)
Classification (cHL)	
Nodular sclerosis	8 (21.6%)
Mixed cellularity	19 (51.3%)
Lymphocyte depleted	1 (2.7%)
Lymphocyte-rich	2 (5.4%)
Unknown	7 (18.9%)

**Table 3 jcm-09-00760-t003:** Cell phenotype in individuals with cHL and controls.

	Pre cHL	cHL	Post cHL	Control Group	*p*		
	N = 6	N = 9	N = 9	N = 9	Pre cHL vs. Control	cHL vs. Control	Post cHL vs. Control
Natural killer cell subsets							
CD56dimCD16-	9.17 [8.43–11.45]	8.99 [7.94–12.29]	6.75 [6.08–9.44]	9.75 [8.18–10.22]	0.413	0.789	0.169
CD56brCD16-	8.53 [6.51–9.08]	5.64 [3.42–7.41]	10.21 [8.06–12.34]	12.24 [8.83–13.76]	*0.050*	0.003	0.290
CD56brCD16+	3.00 [2.14–4.30]	5.24 [3.61–7.15]	4.04 [3.52–4.72]	2.31 [1.24–3.59]	0.224	0.010	0.019
CD56dimCD16+	56.28 [54.58–60.99]	52.94 [47.37–57.05]	52.24 [45.17–61.00]	56.77 [48.15–61.65]	0.866	0.258	0.495
CD56-CD16+	21.55 [16.24–27.56]	25.98 [23.85–27.66]	26.41 [17.53–31.57]	21.46 [17.69–26.52]	0.968	0.161	0.278
Inhibitory CD94+	22.03 [18.11–27.72]	20.36 [17.96–24.46]	20.43 [16.19–22.75]	17.89 [14.05–19.06]	0.088	0.040	0.077
Activating NKp46+	7.87 [6.51–10.63]	8.37 [5.87–9.96]	6.74 [5.98–9.19]	11.24 [9.04–12.16]	0.076	0.031	0.012
Activating NKp30+	1.06 [0.86–1.25]	0.98 [0.78–1.28]	1.14 [0.81–1.42]	1.36 [0.97–1.57]	0.087	0.052	0.234
Activating NKG2D+	1.82 [1.48–2.04]	1.85 [1.47–2.22]	1.74 [1.46–2.14]	2.23 [1.89–2.54]	0.055	0.083	0.077
Lymphocyte B cell subsets							
Resting memory	25.83 [13.92–32.90]	32.56 [24.37–39.91]	32.42 [24.16–35.01]	22.88 [17.07–28.25]	0.689	0.048	0.139
Activated memory	27.58 [18.33–32.22]	16.95 [11.14–22.73]	22.57 [20.20–27.31]	26.15 [19.10–32.11]	0.823	0.018	0.518
Naive	29.34 [20.56–38.41]	40.50 [28.62–46.21]	31.58 [24.65–34.01]	31.02 [28.97–40.52]	0.571	0.347	0.135
Tissue-like memory	14.79 [9.25–23.87]	9.90 [4.47–15.11]	12.24 [10.26–20.16]	10.36 [6.35–18.98]	0.324	0.477	0.702
Immature/Transitional	2.27 [1.41–3.41]	1.17 [0.75–3.14]	1.98 [1.55–3.19]	1.68 [1.22–2.60]	0.456	0.993	0.318
Plasmablast	0.74 [0.53–1.15]	0.11 [0.07–0.80]	0.67 [0.14–0.77]	0.14 [0.09–0.79]	0.224	0.745	0.304
Lymphocyte CD4 T cell subsets							
Naive	18.60 [10.94–21.02]	22.82 [16.59–27.61]	18.57 [10.33–19.85]	17.55 [11.62–23.68]	0.880	0.155	0.496
Central memory	30.20 [26.84–38.63]	43.48 [38.52–46.08]	38.36 [30.99–40.81]	35.98 [24.26–45.45]	0.558	0.072	0.593
Effector memory	47.90 [40.16–51.80]	33.48 [24.14–39.56]	40.84 [39.26–53.19]	41.51 [30.42–63.57]	0.827	0.050	0.945
TemRA+	3.40 [1.91–5.87]	1.36 [0.25–2.80]	1.80 [0.34–2.32]	1.24 [0.80–3.17]	0.067	0.582	0.526
CD4 T cell activation	4.66 [3.57–5.63]	4.86 [3.37–5.31]	4.21 [3.85–5.69]	3.45 [2.67–6.34]	0.661	0.897	0.684
CD4 T cell exhaustion	47.79 [43.14–49.37]	47.98 [42.66–52.52]	51.32 [47.98–58.63]	53.82 [37.06–56.86]	0.636	0.958	0.369
Lymphocyte CD8 T cell subsets							
Naive	21.09 [16.79–24.65]	17.73 [14.73–19.86]	21.33 [13.89–27.38]	18.84 [13.12–24.70]	0.762	0.952	0.612
Central memory	12.88 [10.01–18.53]	24.70 [20.29–35.24]	17.41 [13.45–21.08]	11.23 [5.46–13.23]	0.353	0.006	0.019
Effector memory	55.40 [47.09–61.27]	41.33 [36.80–52.43]	55.32 [49.14–58.14]	58.94 [53.39–65.41]	0.481	0.011	0.225
TemRA+	9.49 [6.45–12.65]	9.31 [6.24–13.91]	8.31 [7.19–11.58]	10.29 [6.59–17.60]	0.838	0.958	0.311
CD8 T cell activation	17.35 [13.11–18.77]	15.54 [14.90–17.89]	13.76 [13.07–18.03]	12.24 [9.48–15.86]	*0.050*	0.031	0.115
CD8 T cell exhaustion	64.40 [61.96–73.23]	65.23 [60.17–68.62]	68.21 [58.43–72.53]	62.99 [55.31–67.54]	0.406	0.532	0.386

**Table 4 jcm-09-00760-t004:** Uni (unadjusted) and multivariate (adjusted) analysis at pre-cHL and cHL diagnosis of biomarkers association with cHL.

	Unadjusted		Adjusted	
	*p*	B (CI 95%)	*p*	B (CI 95%)
Pre-cHL				
miR-16	0.044	0.693 (0.485–0.991)		
miR-20a	0.011	3.272 (1.307–8.192)	0.049	2.784 (1.007–7.700)
miR-21	0.009	1.564 (1.117–2.189)	0.035	1.478 (1.028–2.124)
sCD27	0.033	1.142 (1.011–1.291)		
sCD30	0.042	4.190 (1.053–16.665)		
cHL diagnosis				
Nadir CD4 count	0.080	0.997 (0.994–1.000)		
11CD4/CD8 ratio	0.059	0.167 (0.026–1.074)		
miR-20a	0.007	2.539 (1.290–4.999)	0.011	4.956 (1.443–17.017)
miR-16	0.011	0.636 (0.450–0.900)		
miR-106a	0.005	0.295 (0.126–0.690)		
miR-21	0.020	1.386 (1.053–1.824)		
miR-324	0.003	2.014 (1.260–3.218)		
miR-185	0.034	0.524 (0.288–0.951)		
miR-223	0.039	0.611 (0.383–0.975)		
sCD14	0.017	1.971 (1.130–3.437)		
sCD27	0.007	1.188 (1.047–1.347)		
sCD30	0.017	3.406 (1.242–9.338)		
sIL2R	0.009	1.762 (1.153–2.694)		

**Table 5 jcm-09-00760-t005:** ROC curve and likelihood ratio analysis at pre-cHL and cHL diagnosis.

	ROC Curve				Likelihood Ratio (LR)				
	AUC	*p*	Error	Range	Cut Off	Sensitivity	Specificity	LR+	LR−
Pre-cHL									
miR-20a	0.762	0.008	0.079	0.607–0.917	−0.375	93.8	57.1	2.19	0.11
miR-21	0.756	0.010	0.085	0.589–0.922	6.171	66.7	71.4	2.33	0.47
Combined									
miR-20a+miR-21a	0.832	0.001	0.070	0.694–0.970	0.462	80.0	85.7	5.6	0.23
cHL diagnosis									
miR-20a	0.754	0.005	0.074	0.608–0.900	−0.375	85.7%	57.1%	2	0.25

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
