# Peer review of "Risk, Diagnostic and Predictor Factors for Classical Hodgkin Lymphoma in HIV-1-Infected Individuals: Role of Plasma Exosome-Derived miR-20a and miR-21"

_jcm, 2020, doi:10.3390/jcm9030760_

Round 1

Reviewer 1 Report

Study is of interst and well peformed, the major limitation is the low number of HIV+ cases. However, HIV+ cHL cases are rare and the control groups adjusted is well designed, Thus I consider data herein reported alotugh in a limited number of cases as of interest for the scientifc community and offer important acknoledgment for HIV model of pathogenesis and inderectly on the overall the role of CD4 sutype population of the CHL pathogenesis.

supplementary figure S1 regarding B lymphocyte, please edited the figure since Cd27/CD20 is a cut figure 

Author Response

REVIEWER 1

Study is of interst and well peformed, the major limitation is the low number of HIV+ cases. However, HIV+ cHL cases are rare and the control groups adjusted is well designed, Thus I consider data herein reported alotugh in a limited number of cases as of interest for the scientifc community and offer important acknoledgment for HIV model of pathogenesis and inderectly on the overall the role of CD4 sutype population of the CHL pathogenesis.

Supplementary figure S1 regarding B lymphocyte, please edited the figure since CD27/CD20 is a cut figure.

RESPONSE

Thank you very much for your comments.

Supplementary figure 1-B lymphocyte: I will try to explain why dot plot CD27/CD20 is a cut figure. Dot plot CD27/CD20 shows the events from quadrant superior-left included in dot plot CD27/CD10 using the same CD27 (that is why is a cut image) and CD20 (a different marker). The other two dot plots from CD27/CD10 quadrant sup-left and inferior-left included CD10, so they are cut images to the right of the axis in CD27/CD10 dot plot.    

Reviewer 2 Report

This manuscript reports on biomarkers analyses in cHL,HIV-positive patients, as compared to HIV-patients that have not developed cancer. In particular, miRNA, soluble cytokines, and phenotype analysis are discussed, both prior to cHL diagnosis and after treatment.       Major findings:
  • English language needs to be revised, in particular in the abstract
  • In the methods section of the abstract, T and NK cells phenotyping is mentioned, but these analyses are not commented in the results section, where only miRNA analyses are discussed.
  • The Authors should further comment on the role of EBV infection and in particular on its interaction with immune system of cART-treated patients
  • Authors state that cHL patients were treated with 6 cycles of ABVD regimen; however, in table 2, when describing the stage of cHL patients at diagnosis, only 19/34 patients have stage III-IV cHL; it should be further explained why this regimen was chosen and whether radiotherapy was added for early stage or bulky disease HL.
  • Authors should specify in the text that phenotype analyses was only carried out in a fraction of the cHL cohort, as clearly seen in table 3; in particular, it seems that only 6/34 patients were analyzed before cHL diagnosis, at diagnosis and after treatment: this should be highlighted since it could also explain lack of statistical significance in many of the comparisons made with no-cHL control group; in fact, also figure 1 and 2 require details on number of patients considered in each analysis, as available in the supplementary table.
  • results about miRs described in extenso in page 14-15 could be significantly shortened, since they are all available in the tables/figures; this would leave more space to comment on significant associations and on impact on clinical outcome
  • Outcome of the cHL patients should be better explained, in particular it is not clear whether death was HL-related, AIDS-related or if death by any cause was considered as an event. Since 4/34 patients have died at a follow up of nearly 1 year after cHL diagnosis, it seems this is an important matter to discuss.
  • Comparison of biomarkers with cHL, non-HIV patients is not available, as well as comparison with general population: this should also be commented in the section regarding the limitations of the study (page 18)
Minor findings:
  • in the introduction section, row 42, it is not clear whether incidence of cHL has risen in the HIV+ population after the introduction of cART, or if the authors simply refer to a comparison with the general population in the same time window
  • figure 1 legend needs to be fixed
  • a few typos should be corrected throughout the text

Author Response

REVIEWER 2

  • English language needs to be revised, in particular in the abstract.

RESPONSE: It has been revised

  • In the methods section of the abstract, T and NK cells phenotyping is mentioned, but these analyses are not commented in the results section, where only miRNA analyses are discussed.

RESPONSE: The most significant results regarding T, B and NK subsets are mentioned in the results section on page 9, lines 164-167; page 14, lines 194-201; and page 15, lines 227-229.

  • The Authors should further comment on the role of EBV infection and in particular on its interaction with immune system of cART-treated patients.

RESPONSE: An additional comment has been added in the Introduction section.

  • Authors state that cHL patients were treated with 6 cycles of ABVD regimen; however, in table 2, when describing the stage of cHL patients at diagnosis, only 19/34 patients have stage III-IV cHL; it should be further explained why this regimen was chosen and whether radiotherapy was added for early stage or bulky disease HL.

RESPONSE: The referee is right. This needs to be further explained. Patients with advanced stages received 6 to 8 cycles of ABVD, while patients with early stages (stages I and II) received combined therapy including 2 cycles of ABVD followed by RT and 2 more cycles of ABVD. This explanation has been added to the Results section.

  • Authors should specify in the text that phenotype analyses was only carried out in a fraction of the cHL cohort, as clearly seen in table 3; in particular, it seems that only 6/34 patients were analyzed before cHL diagnosis, at diagnosis and after treatment: this should be highlighted since it could also explain lack of statistical significance in many of the comparisons made with no-cHL control group; in fact, also figure 1 and 2 require details on number of patients considered in each analysis, as available in the supplementary table.

RESPONSE: The number of plasma and PBMC samples is added in the Materials and Methods section with a mention to see Supplementary Table S1.

Supplementary Table S1 has been slightly modified: Total samples of plasma and PBMCs have been added and a mistake has been fixed (the number of plasma single samples in post cHL (after treatment) was set by 6 patients instead of 7 patients).

Table 3 has been modified in the number of patients analyzed at Post cHL, N=9 instead of N=6.

Figures 1 and 2 show the number of patients with plasma samples, i.e, 15 patients at Pre cHL, 25 patients at cHL, and 18 patients at Post cHL, as mentioned in Materials and Methods and in Supplementary Table S1.

  • Results about miRs described in extenso in page 14-15 could be significantly shortened, since they are all available in the tables/figures; this would leave more space to comment on significant associations and on impact on clinical outcome.

RESPONSE: Thank you for the advice.

  • Outcome of the cHL patients should be better explained, in particular it is not clear whether death was HL-related, AIDS-related or if death by any cause was considered as an event. Since 4/34 patients have died at a follow up of nearly 1 year after cHL diagnosis, it seems this is an important matter to discuss.

RESPONSE: The referee is right. All four deaths were cHL-related. This has been included in the text.

  • Comparison of biomarkers with cHL, non-HIV patients is not available, as well as comparison with general population: this should also be commented in the section regarding the limitations of the study (page 18).

RESPONSE: This limitation has been added in the Discussion section.

Minor findings:

  • In the introduction section, row 42, it is not clear whether incidence of cHL has risen in the HIV+ population after the introduction of cART, or if the authors simply refer to a comparison with the general population in the same time window.

RESPONSE: This paragraph has been modified trying to clarify both concepts; the evolution of the incidence of cHL among HIV-1-infected individuals, and the incidence of HIV-related cHL compared to healthy individuals.  

  • figure 1 legend needs to be fixed

RESPONSE: It has been fixed.

  • a few typos should be corrected throughout the text

RESPONSE: Text has been sent out for grammar edition.